# Timing of eye removal influences low-vision quality of life and self-perception of facial appearance in people with one eye

**Aysha N. Kinakool**(iD)**, Stefania S. Moro, Jennifer K. E. Steeves**(iD)*

Department of Psychology, Centre for Vision Research, and Centre for Integrative and Applied Neuroscience, York University, Toronto, Canada

* steeves@yorku.ca

## Abstract

### Purpose

The loss of one eye early compared to late in life coincides with critical periods of postnatal visual development and is associated with sensory adaptations which may lead to different psychosocial experiences throughout life. This study investigates the impact of age at enucleation on psychosocial health factors, namely low-vision quality of life, self-perception of facial appearance, anxiety, and depression.

### Methods

Twenty-two participants with early or late eye removal and twenty-seven binocular viewing controls completed the Low-Vision Quality of Life questionnaire, Facial Appearance sub-scale of the Negative Physical Self-Scale and the Hospital Anxiety and Depression scale.

### Results

Low-vision quality of life was lower for people who had their eye removed late in life compared to binocular viewing controls. Further, within this group, lower self-perception of facial appearance was associated with lower low-vision quality of life. Moreover, experience with binocularity and time since enucleation were not associated with psychosocial health factors.

### Conclusion

Developmental factors rather than experiential factors appear to influence psychosocial health since binocular experience and time since enucleation were not related to any psychosocial factor. These findings suggest that people who have had their eye removed early in life, during the critical periods of visual development may benefit from better psychosocial outcomes.

**Data availability statement:** Data are available on an open-access data repository (https://doi.org/10.5683/SP3/ZBFRPC).

**Funding:** This research was funded by Canada First Research Excellence Fund: Vision Science to Application (VISTA) (CFREF-2015-00013); Natural Sciences and Engineering Research Council of Canada (NSERC) (327588); Canada Foundation for Innovation (CFI) (12807) to author JKES. The funders had no role in study design, data collection and analysis, decision to publish, or preparation of the manuscript.

**Competing interests:** The authors have declared that no competing interests exist.

## Introduction

At birth, the visual system is physiologically and functionally not fully developed and balanced binocular input allows for typical postnatal maturation [1]. Multiple overlapping critical periods exist that determine the development of typical visual function and visual maturation can extend into our third decade of life for some functions (e.g., face recognition) [2]. Disruption of visual input during this maturation period can lead to anatomical changes along different structures in the visual pathway, changes in cortical function, as well as changes in visual and audiovisual behaviour [1]. The timing of visual disruption can have differential effects on visual maturation.

A unique model for examining the consequences of the loss of binocularity during the critical periods of visual development is unilateral eye enucleation [3]. Unilateral eye enucleation is the surgical removal of one eye due to, for example ocular malignancy, traumatic injury, or eye disease [4]. This yields complete deafferentation of one channel of visual input to the brain unlike other forms of partial visual deprivation such amblyopia which frequently has remaining anomalous visual signals from the affected eye [3,5]. There is a growing body of behavioural and neuroimaging research showing physiological and functional adaptations that accommodate when one eye is lost early in life (e.g., reviewed in [3,5]). Early eye enucleation has been shown to be associated with changes in subcortical and cortical structures [6], visual, auditory, and audiovisual brain function and behaviour [6–12].

These adaptations do not appear when one eye is removed late in life. For example, contrast sensitivity shows enhancements for early but not late eye enucleation [13], indicating a developmental relationship between visual ability and the timing of the loss of one eye. Removing one eye late in life is associated with more reduction in volume compared to losing one eye early in life in the lateral geniculate nucleus (the visual relay centre of the thalamus) [7]. Adaptations in other sensory systems have also been observed across sensory modalities. People who have had one eye removed early in life have reduced susceptibility to audiovisual illusions such as the McGurk effect [8–11] unlike those who had one eye removed late in life and binocular viewing control participants [14]. This indicates that binocular experience and its relation to critical periods of visual development influences cortical reorganization and behaviour differently in people who have had one eye removed early compared to late in life. Considering the differential effects of early compared to late eye enucleation, it is plausible that timing of eye enucleation not only impacts the brain and its sensory abilities but could also influence psychosocial visual experience with the world and ultimately quality of life.

Previous research has shown that removing an eye is related to changes in psychosocial experience and quality of life. While some studies have shown people who have had an eye removed have higher or a similar quality of life to healthy controls [15–18], there are also reports of lower quality of life [19–22]. Moreover, people with one eye have reported feelings of social anxiety and avoid social situations associated with their altered appearance and negative feelings about how they are viewed [23]. This avoidance and social anxiety can lead to feelings of anger, depression, and isolation

[24]. Eye enucleation in general is associated with increased anxiety and depression and decreased positive self-perception of facial appearance [25–27]. For instance, higher levels of anxiety and depression are associated with lower vision-related quality of life and with greater concerns about facial appearance [27,28]. Higher anxiety is associated with decreased appearance related social function but is not associated with vision-related quality of life [25]. Some have found anxiety and depression levels are elevated but nonetheless within the typical range for the general population [23] while others have found increased prevalence of anxiety and depression compared to the general population [29]. The variability across studies may be a reflection of differences between having one's eye removed early compared to late in life.

Since there is some evidence that removing one eye is related to changes in psychosocial factors and that the timing of visual disruption can have differential effects on visual system maturation, it is plausible that timing of enucleation may be an important factor in determining psychosocial outcomes. The aim of this study was to investigate the impact of age at enucleation and psychosocial health, namely low-vision quality of life (LVQoL), self-perception of facial appearance, anxiety, and depression. We hypothesized that because adults who have had their eye removed early in life have experienced different physiological and behavioural adaptions compared to adults who have had their eye removed late in life, these adaptations may be related to a different psychosocial trajectory.

## Methods

### Participants

**Participants with early monocular enucleation (E-ME).** Thirteen adult participants who had undergone early monocular enucleation participated, mean (SD) = 45 (10) years. All E-ME participants had undergone unilateral eye enucleation (8 right and 5 left eye) due to retinoblastoma (a childhood cancer of the retina, n = 11), traumatic injury (n = 1), or eye disease (n = 1). Age at enucleation ranged from 12–60 months, mean (SD) = 20 (13) months.

**Participants with late monocular enucleation (L-ME).** Nine adult participants who had undergone late monocular eye enucleation participated, mean (SD) = 51 (10) years. All L-ME participants had been unilaterally enucleated (2 right eye and 7 left eye) due to ocular malignancy (n = 1), traumatic injury (n = 7), or other (n = 1). Age at enucleation ranged from 84–744 months, mean (SD) = 335 (223) months.

**Binocular viewing participants (BV).** The control group consisted of twenty-seven binocularly intact participants, mean (SD) = 47 (11) years.

Participant recruitment and data collection were conducted in three testing intervals – 1st November 2021–13th February 2022; 14th April 2022–16th January 2023; and 7th November 2023–1st January 2024. Written consent was obtained from all participants. All participants (E-ME, L-ME, and BV) reported normal or corrected-to-normal vision (in the remaining eye for participants with one eye). Participants were excluded if they had any other eye diseases, psychiatric, or neurological conditions. The study was approved by the York University Office of Research Ethics (Ethics Certificate: e2021-085) and adhered to the tenets of the Declaration of Helsinki. Participant demographics are presented in Table 1.

### Procedure

A self-administered cross-sectional online survey was presented using Qualtrics (Qualtrics XM, Provo, Utah, USA). Four indicators related to psychosocial health were measured: low-vision quality of life (LVQoL), self-perception of facial appearance, anxiety, and depression. LVQoL was assessed using the Low-Vision Quality of Life questionnaire which examines four areas: 1. Distance vision, mobility, and lighting; 2. Adjustments; 3. Reading and fine work; and 4. Activities of daily living [30]. A higher LVQoL score indicates that an individual has a higher quality of life in relation to their vision. Self-perception of facial appearance was assessed using the Facial Appearance sub-scale of the Negative Physical Self-Scale where higher scores indicate a higher self-perception of one's facial appearance [31]. Finally, anxiety and depression were assessed using the Hospital Anxiety and Depression scale where higher scores indicate higher levels of anxiety and depression [32].

**Table 1. Participant Demographics.**

| Participant ID | Group | Age (years) | Sex | Ethnicity | Eye Removed | Age at Enucleation (months) |
|---|---|---|---|---|---|---|
| E001 | E-ME | 37 | F | White | Left | 12 |
| E002 | E-ME | 48 | F | Asian | Left | 12 |
| E003 | E-ME | 34 | F | White | Right | 12 |
| E004 | E-ME | 51 | M | White/ South Asian | Left | 24 |
| E005 | E-ME | 41 | M | White | Left | 12 |
| E006 | E-ME | 40 | F | White | Right | 12 |
| E007 | E-ME | 42 | M | White | Left | 12 |
| E008 | E-ME | 56 | F | White | Right | 24 |
| E009 | E-ME | 56 | F | White | Right | 24 |
| E010 | E-ME | 37 | F | White | Right | 24 |
| E011 | E-ME | 41 | M | White | Right | 12 |
| E012 | E-ME | 68 | M | White | Right | 60 |
| E013 | E-ME | 34 | M | White | Right | 24 |
| L001 | L-ME | 53 | F | White/ Asian | Right | 624 |
| L002 | L-ME | 62 | F | White | Left | 744 |
| L003 | L-ME | 59 | F | White | Left | 312 |
| L004 | L-ME | 55 | F | White | Left | 84 |
| L005 | L-ME | 49 | M | West Asian | Left | 360 |
| L006 | L-ME | 56 | M | White | Left | 144 |
| L007 | L-ME | 58 | M | White | Left | 216 |
| L008 | L-ME | 31 | F | White | Right | 372 |
| L009 | L-ME | 39 | F | White | Left | 156 |
| C001 | BV | 34 | M | South Asian | NA | NA |
| C002 | BV | 30 | F | South Asian | NA | NA |
| C003 | BV | 61 | M | South Asian | NA | NA |
| C004 | BV | 38 | F | South Asian | NA | NA |
| C005 | BV | 59 | F | White | NA | NA |
| C006 | BV | 58 | M | White | NA | NA |
| C007 | BV | 50 | F | White | NA | NA |
| C008 | BV | 61 | F | White | NA | NA |
| C009 | BV | 66 | F | White | NA | NA |
| C010 | BV | 57 | F | White | NA | NA |
| C011 | BV | 59 | F | White | NA | NA |
| C012 | BV | 56 | F | White | NA | NA |
| C013 | BV | 32 | M | White | NA | NA |
| C014 | BV | 55 | F | White | NA | NA |
| C015 | BV | 42 | M | White | NA | NA |
| C016 | BV | 42 | F | White | NA | NA |
| C017 | BV | 57 | F | White | NA | NA |
| C018 | BV | 43 | M | White | NA | NA |
| C019 | BV | 32 | M | South Asian | NA | NA |
| C020 | BV | 37 | M | South Asian | NA | NA |
| C021 | BV | 48 | F | Mixed | NA | NA |
| C022 | BV | 41 | M | White | NA | NA |
| C023 | BV | 36 | F | South Asian | NA | NA |
| C024 | BV | 54 | M | White | NA | NA |

*(Continued)*

**Table 1.** (Continued)

| Participant ID | Group | Age (years) | Sex | Ethnicity | Eye Removed | Age at Enucleation (months) |
|---|---|---|---|---|---|---|
| C025 | BV | 37 | M | Mixed | NA | NA |
| C026 | BV | 32 | M | White | NA | NA |
| C027 | BV | 52 | M | White | NA | NA |

E-ME: Early Monocular Enucleation; L-ME: Late Monocular Enucleation; BV: Binocular Viewing.

## Results

### Psychosocial factors across groups

Non-parametric Kruskal Wallis tests were conducted for non-normal data between groups (E-ME, L-ME, BV) for each psychosocial factor (LVQoL, self-perception of facial appearance, anxiety, and depression). Group differences were found for LVQoL, $\chi^2$ (2, 49) = 14.38, p = 0.001, $\eta_p^2$ = 0.230. Post-hoc pairwise comparisons using Dunn-Bonferroni correction indicated LVQoL for L-ME was lower compared to BV controls (p = 0.001) (Fig 1A). There was no difference in self-perception of facial appearance, $\chi^2$ (2, 49) = 3.48, p = 0.176, $\eta_p^2$ = 0.068 (Fig 1B). There was no difference in anxiety scores, $\chi^2$ (2, 49) = 3.26, p = 0.196, $\eta_p^2$ = 0.064 (Fig 1C). Finally, there was no difference in depression scores, $\chi^2$ (2, 49) = 0.13, p = 0.936, $\eta_p^2$ = 0.003 (Fig 1D). Fig 1 illustrates the group differences for each psychosocial factor.

### Self-perception of facial appearance and psychosocial factors

Spearman correlations were conducted comparing psychosocial factors (LVQoL, anxiety, and depression) and self-perception of facial appearance for each group (E-ME, L-ME, BV). There was a strong positive correlation where lower self-perception of facial appearance was associated with lower LVQoL in the L-ME group, $r_s(7)$ = 0.77, p = 0.016 (Fig 2A). There was no relation between self-perception of facial appearance and LVQoL for E-ME, $r_s(11)$ = 0.32, p = 0.281 or BV controls, $r_s(25)$ = -0.26, p = 0.188.

Spearman correlations were conducted comparing self-perception of facial appearance and anxiety and indicated an inverse relation where low anxiety was associated with high self-perception of facial appearance in the L-ME ($r_s(7)$ = -0.78, p = 0.014), E-ME ($r_s(11)$ = -0.78, p = 0.002), and BV groups ($r_s(25)$ = -0.38, p = 0.050) (Fig 2B).

Finally, Spearman correlations were conducted comparing self-perception of facial appearance and depression and indicated a strong inverse relation where low depression was associated with high self-perception of facial appearance in the L-ME ($r_s(7)$ = -0.90, p < 0.001), E-ME ($r_s(11)$ = -0.75, p = 0.003), and BV groups ($r_s(25)$ = -0.48, p = 0.010) (Fig 2C).

### Binocular experience and psychosocial factors

In order to explore whether binocular experience, independent of age, impacts psychosocial factors we examined the relation between the number of months a person lived with binocular vision (before monocular enucleation) and psychosocial factors (LVQoL, self-perception of facial appearance, anxiety, and depression). A Mann Whitney U test comparing binocular experience between monocular enucleation groups (E-ME and L-ME) indicated that the E-ME group had less binocular experience compared to the L-ME group, $U$ = -91, p < 0.001, r = -2.13. Binocular viewing control participants were not included in these analyses as binocular experience is simply a reflection of their age.

Spearman correlation analyses were conducted to investigate the relation between binocular experience and psychosocial factors (LVQoL, self-perception of facial appearance, anxiety, and depression). There was no relation between binocular experience and LVQoL, $r_s(20)$ = -0.32, p = 0.150 (Fig 3A); self-perception of facial appearance, $r_s(20)$ = 0.12, p = 0.593 (Fig 3B); anxiety, $r_s(20)$ = -0.08, p = 0.739 (Fig 3C); or depression, $r_s(20)$ = -0.01, p = 0.956 (Fig 3D).

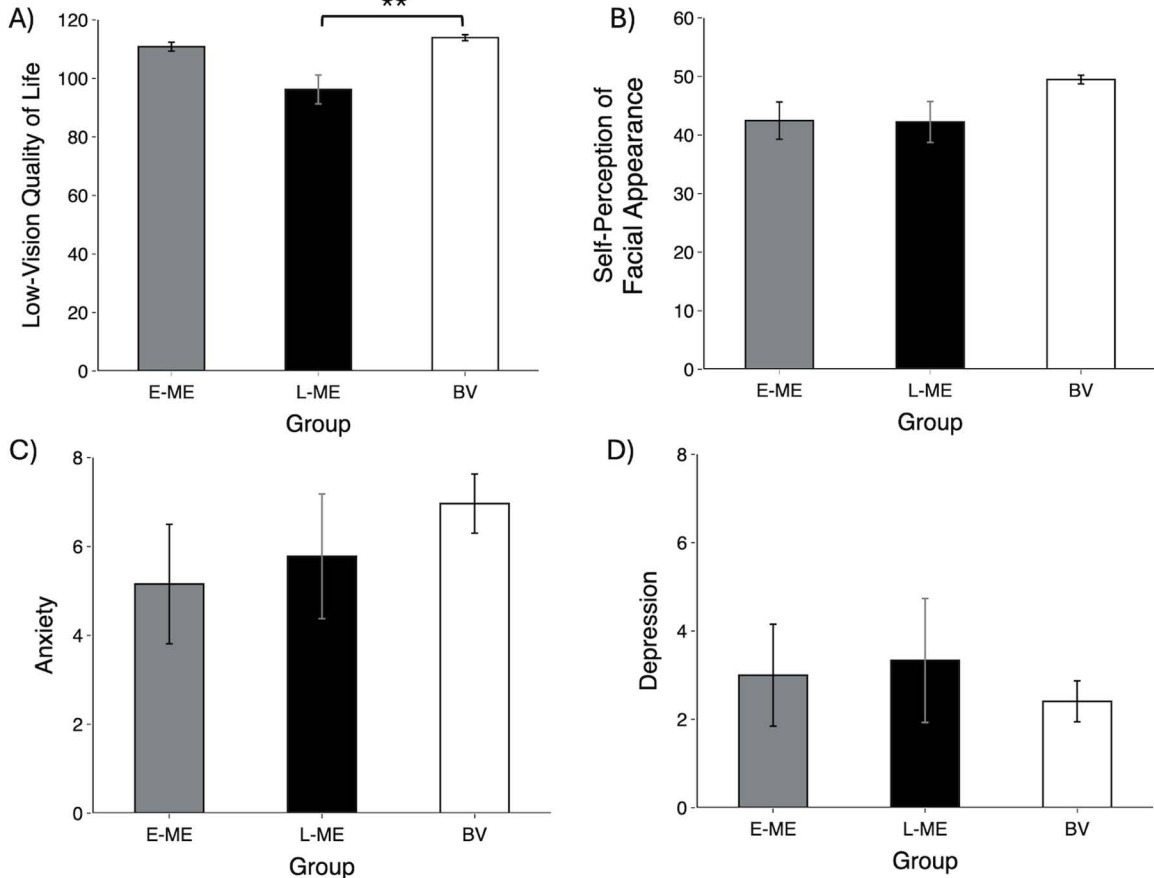

**Fig 1. Mean scores of psychosocial factors across groups;** A) Low-vision quality of life (LVQoL), B) Self-perception of facial appearance, C) Anxiety, and D) Depression. LVQoL of L-ME is lower than BV group. Error bars represent standard error of the mean (SEM). E-ME = Participants with early enucleation (grey), L-ME = Participants with late enucleation (black), BV = Binocular viewing (white). ** p < 0.01.

### Time since enucleation and psychosocial factors

In order to explore whether time since enucleation impacts psychosocial factors, we examined the relation between the number of months a person has lived since enucleation and psychosocial factors (LVQoL, self-perception of facial appearance, anxiety, and depression). A Mann Whitney U test indicated that the E-ME group lived a longer time since enucleation compared to the L-ME group, $U = 2$, $p = 0.023$, $r = -0.80$.

Spearman correlation analyses showed no relation between time since enucleation and LVQoL, $r_s(20) = 0.30$, $p = 0.181$ (Fig 4A); self-perception of facial appearance, $r_s(20) = 0.14$, $p = 0.532$ (Fig 4B); anxiety, $r_s(20) = -0.35$, $p = 0.108$ (Fig 4C); or depression, $r_s(20) = -0.13$, $p = 0.571$ (Fig 4D).

### Discussion

A growing body of research has established that removing one eye early in life affords different cortical, functional, and behavioural adaptations compared to people who have had their eye removed late in life, as adults. It is possible that these developmental adaptations contribute to improving future psychosocial experiences related to vision health. Our findings indicate that people who had their eye removed late in life have lower LVQoL compared to binocular viewing control participants. In addition, this finding was associated with self-perception of facial appearance, where lower

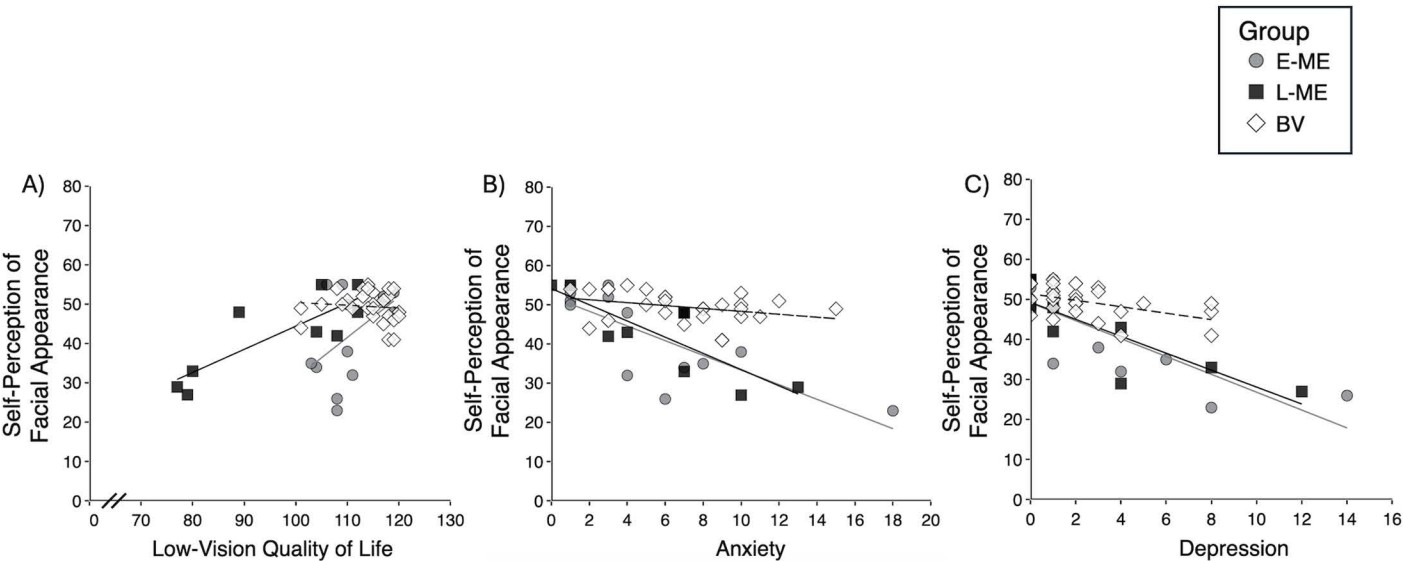

**Fig 2. Scatterplots depicting the relation between self-perception of facial appearance and A)** Low-vision quality of life (measured using LVQoL), **B)** Anxiety (measured using Hospital Anxiety and Depression Questionnaire), and **C)** Depression (measured using Hospital Anxiety and Depression Questionnaire). E-ME = Participants with early enucleation (grey circles), L-ME = Participants with late enucleation (black squares), BV = Binocular viewing (white diamonds).

self-perception of facial appearance is associated with lower LVQoL in this group. Overall, low levels of anxiety and depression were associated with higher perception of facial appearance regardless of whether a participant had an eye removed. Binocular experience and time since enucleation were not related to any psychosocial factor.

People who had one eye removed late in life reported lower LVQoL compared to binocular viewing control participants, unlike people who had an eye removed early in life. In addition, self-perception of facial appearance was also a factor contributing to lower LVQoL but only for participants who had their eye removed late in life. This indicates that there is a distinction between having your eye removed early compared to late in life. Participants in the early monocular enucleation group had their eye removed before 5 years of age where visual disruption occurred during the early critical periods of vision development. It has been established that there are physiological and functional adaptations when one eye is removed early in life (reviewed in [3,5]). These adaptations may impact experience and psychological outlook more broadly in this early monocular enucleation group. This seems plausible since our participants who had an eye removed early in life did not differ on psychosocial factors compared to binocularly intact controls.

In contrast, the average age of eye enucleation of participants who had their eye removed late in life was at approximately 28 years of age – well outside the critical periods of post-natal brain maturation and vision development. A growing body of research shows that monocular enucleation late in life, such as this, is not associated with the same physiological and functional adaptations observed when one eye is removed early in life [7,14]. It appears that the timing of interruption to visual input during post-natal development is a key explanation for this finding. In order to investigate whether this finding was related to developmental factors (such as timing of visual input disruption during development) compared to experiential factors, we assessed time since enucleation (number of months that have passed since removal of one eye) and binocular experience (number of months with binocular vision prior to eye removal). However, we found no relation to the amount of time that had passed since eye enucleation and no relation to binocular experience (time lived with binocular vision). This suggests a greater influence of timing of eye enucleation during development rather than experience with

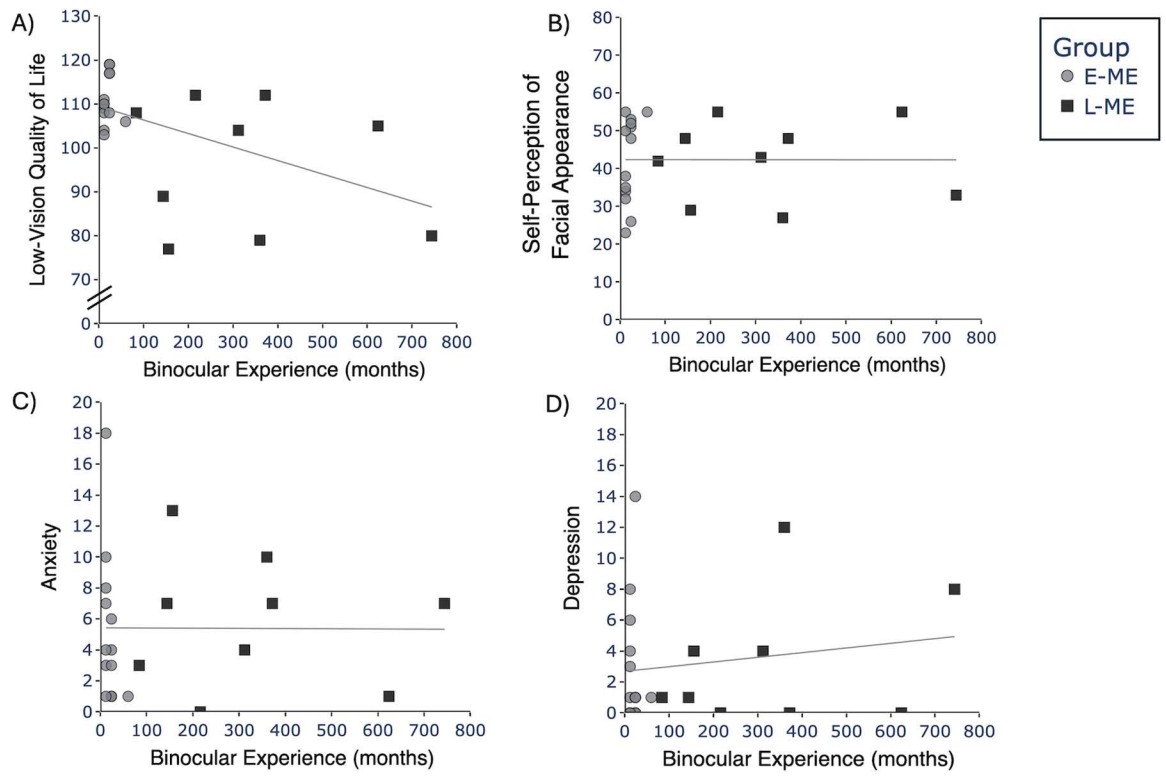

**Fig 3. Scatterplots depicting the relation between binocular experience and A)** Low-vision quality of life (LVQoL), **B)** Self-perception of facial appearance, **C)** Anxiety, and **D)** Depression across groups. E-ME = Participants with early enucleation (grey circles), L-ME = Participants with late enucleation (black squares).

only one eye. Previous studies investigating the impact of removing one eye on psychosocial factors have not distinguished between those who had one eye removed early compared to late in life [16,17,20,25,27,28].

Overall, the current study provides novel insights into factors that contribute to the psychosocial health of people who have had one eye removed, differentiating between those who have had their eye removed early compared to late in life. This may suggest that people who have had one eye removed early in life, during the critical periods of visual development, may benefit from better psychosocial outcomes such as higher LVQoL. Given the nature of this qualitative study and the challenges associated with recruiting patient groups, limitations can be noted. In comparison to qualitative studies of the general population, our study had a relatively small patient sample size (n = 22). The rare nature of eye enucleation poses challenges to recruitment. Additionally, our study was conducted online which limits our ability to provide controlled testing conditions. However, online studies are ideal for rare patient groups who may have accessibility and geographical constraints. It would be valuable for future studies to consider objective measures of participants' eye prostheses in the event that prosthetic quality is a co-factor for psychosocial health.

In conclusion, our study indicates that developmental factors rather than experiential factors influence psychosocial health in people with one eye. This is reflected in our findings which show that experience with binocularity and time since enucleation are not related to psychosocial health factors. Instead, it is likely that the physiological and functional adaptations, driven by neuroplasticity during the critical periods of visual development, may provide a quality-of-life advantage for individuals who had an eye removed early in life. Our findings underscore the importance of age at which enucleation occurs when considering psychosocial health in people who have had one eye removed in both clinical practice and future research.

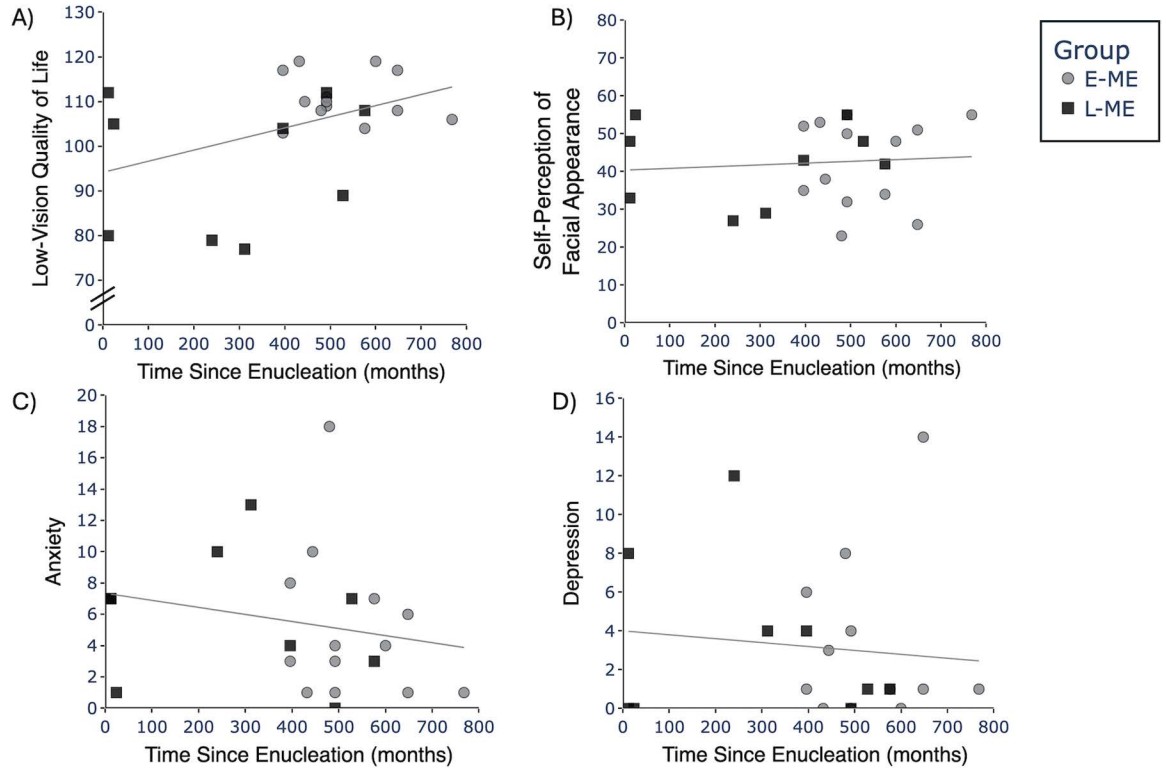

**Fig 4. Scatterplots depicting the relation between time since enucleation and A)** Low-vision quality of life (LVQoL), **B)** Self-perception of facial appearance, **C)** Anxiety, and **D)** Depression. E-ME = Participants with early enucleation (grey circles), L-ME = Participants with late enucleation (black squares).

## Acknowledgments

The authors would like to thank all the participants for their time in completing this study.

## Author contributions

**Conceptualization:** Aysha N. Kinakool, Stefania S. Moro, Jennifer K. E. Steeves.

**Formal analysis:** Aysha N. Kinakool, Stefania S. Moro, Jennifer K. E. Steeves.

**Funding acquisition:** Jennifer K. E. Steeves.

**Supervision:** Jennifer K. E. Steeves.

**Writing – original draft:** Aysha N. Kinakool.

**Writing – review & editing:** Aysha N. Kinakool, Stefania S. Moro, Jennifer K. E. Steeves.

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
