## [Decision Letter · Decision Letter 0]

18 Mar 2025

PONE-D-24-53633Timing of eye removal influences low-vision quality of life and self-perception of facial appearance in people with one eyePLOS ONE

Dear Dr. Steeves,

Thank you for submitting your manuscript to PLOS ONE. After careful consideration, we feel that it has merit but does not fully meet PLOS ONE’s publication criteria as it currently stands. Therefore, we invite you to submit a revised version of the manuscript that addresses the points raised during the review process.

We look forward to receiving your revised manuscript.

Kind regards,

Mohamed Fawzy Mohamed Hamed Shehata

Academic Editor

PLOS ONE

Journal Requirements:

“This research was supported by grants from Canada First Research Excellence Fund: Vision Science to Application (VISTA) (CFREF-2015-00013); Natural Sciences and Engineering Research Council of Canada (NSERC) (327588); Canada Foundation for Innovation (CFI) (12807).”

Additional Editor Comments:

Please try to addresses the minor comments from reviewers

Reviewers' comments:

Reviewer's Responses to Questions

**Comments to the Author**

1. Is the manuscript technically sound, and do the data support the conclusions?

Reviewer #1: Yes

Reviewer #2: Yes

2. Has the statistical analysis been performed appropriately and rigorously? 

Reviewer #1: Yes

Reviewer #2: Yes

3. Have the authors made all data underlying the findings in their manuscript fully available?

Reviewer #1: Yes

Reviewer #2: Yes

4. Is the manuscript presented in an intelligible fashion and written in standard English?

Reviewer #1: Yes

Reviewer #2: Yes

5. Review Comments to the Author

Reviewer #1: The paper is highly informative and meets the criteria for publication in your journal. It provides valuable insights that will be beneficial for future scientists. Given its significance, I strongly recommend its consideration for publication.

Reviewer #2: The manuscript addresses important issue related to mononuclear vision and the effect of eye enucleation timing on psychological and social aspects of life. The manuscript is nicely written and advances the knowledge related to post enucleation experience in humans. Only one point to be considered by authors, please provide full demographic data of the study subjects (sex, ethnicity, etc.). This will give full clarity to the research and enhance it global exposure.

6. PLOS authors have the option to publish the peer review history of their article (what does this mean? ). If published, this will include your full peer review and any attached files.

**Do you want your identity to be public for this peer review?** For information about this choice, including consent withdrawal, please see our Privacy Policy .

Reviewer #1: No

Reviewer #2: No

---

## [Author Response · Author response to Decision Letter 1]

7 Apr 2025

Response to Reviewers:

Reviewer #1: The paper is highly informative and meets the criteria for publication in your journal. It provides valuable insights that will be beneficial for future scientists. Given its significance, I strongly recommend its consideration for publication.

Response: Thank you for your review!

Reviewer #2: The manuscript addresses important issue related to mononuclear vision and the effect of eye enucleation timing on psychological and social aspects of life. The manuscript is nicely written and advances the knowledge related to post enucleation experience in humans. Only one point to be considered by authors, please provide full demographic data of the study subjects (sex, ethnicity, etc.). This will give full clarity to the research and enhance it global exposure.

Response: We have added a table with demographic data to our manuscript.

---

## [Editor Report · Decision Letter 1]

11 Apr 2025

Timing of eye removal influences low-vision quality of life and self-perception of facial appearance in people with one eye

PONE-D-24-53633R1

Dear Dr. Steeves,

We’re pleased to inform you that your manuscript has been judged scientifically suitable for publication and will be formally accepted for publication once it meets all outstanding technical requirements.

Kind regards,

Mohamed Fawzy Mohamed Hamed Shehata

Academic Editor

PLOS ONE
---

## [Editor Report · Acceptance letter]

PONE-D-24-53633R1

PLOS ONE

Dear Dr. Steeves,

I'm pleased to inform you that your manuscript has been deemed suitable for publication in PLOS ONE. Congratulations! Your manuscript is now being handed over to our production team.

Kind regards,

on behalf of

Professor Mohamed Fawzy Mohamed Hamed Shehata

Academic Editor

PLOS ONE